# NEURAL POLICY ITERATION FOR STOCHASTIC OPTIMAL CONTROL: A PHYSICS-INFORMED APPROACH

## ABSTRACT

We propose a physics-informed neural network policy iteration (PINN-PI) framework for solving stochastic optimal control problems governed by second-order Hamilton–Jacobi–Bellman (HJB) equations. At each iteration, a neural network is trained to approximate the value function by minimizing the residual of a linear PDE induced by a fixed policy. This linear structure enables systematic $L^2$ error control at each policy evaluation step, and allows us to derive explicit Lipschitz-type bounds that quantify how value gradient errors propagate to the policy updates. This interpretability provides a theoretical basis for evaluating the quality of policy during training. Our method extends recent deterministic PINN-based approaches to stochastic settings, inheriting the global exponential convergence guarantees of classical policy iteration under mild conditions. We demonstrate the effectiveness of our method on several benchmark problems, including stochastic cartpole, pendulum problems and high-dimensional linear quadratic regulation (LQR) problems in up to 20D.

## 1 INTRODUCTION

Solving infinite-horizon stochastic optimal control problems requires computing the value function, which satisfies a nonlinear Hamilton–Jacobi–Bellman (HJB) partial differential equation (PDE). In high dimensions, traditional numerical methods become intractable due to the curse of dimensionality. While Howard's policy iteration (PI) Howard (1960); Puterman & Brumelle (1979); Puterman (1981) provides a theoretically grounded and convergent scheme, each iteration requires solving a linear PDE, which becomes the computational bottleneck in practice.

Recent theoretical works have extended PI to various settings: deterministic control Tang et al. (2025), stochastic control under viscosity solutions Jacka & Mijatović (2017); Kerimkulov et al. (2020), and entropy-regularized (exploratory) formulations Tran et al. (2025b); Huang et al. (2025). These developments underscore the robustness of the PI framework, but also highlight the need for scalable solvers that can handle high-dimensional PDEs with theoretical guarantees.

In this work, we propose a mesh-free, physics-informed policy iteration framework for solving stochastic optimal control problems governed by second-order HJB equations. Our method integrates classical PI with physics-informed neural networks (PINNs): at each iteration, the value function is approximated by a neural network trained to minimize the residual of the linear PDE associated with a fixed policy. By fixing the policy at each iteration, we obtain a linear PDE for the value function, which contrasts with the fully nonlinear HJB equation. This linearity enables the use of classical energy estimates to control the $L^2$ error, which would be difficult to establish under direct optimization of the full HJB.

Unlike model-free reinforcement learning or trajectory-based PINN approaches, our method directly targets the PDE structure of the control problem. This yields both theoretical interpretability and numerical scalability. In particular, we show that value gradient error controls policy error through a Lipschitz-type bound, enabling policy quality monitoring throughout training.

We validate our approach on high-dimensional stochastic control tasks, including LQR, pendulum, and cartpole problems. Results confirm that our method retains the convergence and stability of classical PI while benefiting from the flexibility of neural PDE solvers. Our main contributions are:

- We propose a physics-informed, mesh-free policy iteration framework for solving high-dimensional stochastic control problems governed by nonlinear HJB equations.
- We establish a rigorous $L^2$ error analysis with a decomposition into iteration error, residual error, and policy mismatch, and prove global exponential convergence under standard assumptions.
- We demonstrate the accuracy and scalability of our approach on a variety of nonlinear stochastic control benchmarks.

## 2 RELATED WORK

**Classical Policy Iteration.** Policy iteration (PI) was first formalized by Howard Howard (1960) and later analyzed in depth by Puterman et al. Puterman & Brumelle (1979), who connected PI to Newton–Kantorovich iterations and established convergence rates. In continuous-time settings, Puterman Puterman (1981) extended these ideas to controlled diffusion processes, showing convergence under appropriate assumptions.

**Viscosity Methods for HJB Equations.** In the context of continuous-time stochastic control, monotone convergence of PI under a weak solution framework has been established Jacka & Mijatović (2017). Slightly later, global exponential convergence using BSDE-based techniques were proposed Kerimkulov et al. (2020). For deterministic control problems, the convergence of PI was analyzed within the framework of viscosity solution Tang et al. (2025).

**Entropy-Regularized and Exploratory Control.** Entropy-regularized HJB equations arise in exploratory control settings, where the optimal policy is stochastic due to the inclusion of an entropy term in the objective. Convergence of policy iteration in this context has been studied extensively Tran et al. (2025a); Huang et al. (2025). In particular, under the assumption that the diffusion coefficient depends weakly (or not at all) on the control variable, geometric convergence has been established.

**Physics-Informed and Neural Approaches.** Physics-informed neural networks (PINNs) Raissi et al. (2019) have emerged as mesh-free alternatives for solving high-dimensional PDEs, offering flexibility and scalability beyond traditional discretization methods. Neural variants of policy iteration have combined these tools with classical control frameworks. One such approach introduces ELM-PI and PINN-PI, which solve linearized PDEs in deterministic control problems and support Lyapunov-based stability verification Meng et al. (2024). Additional extensions include nonconvex formulations Yang et al. (2025), operator-learning-based architectures Lee & Kim (2025), and reinforcement learning methods that integrate differentiable physics or PDE solvers into model-based pipelines Ramesh & Ravindran (2023); Mukherjee & Liu (2023).

Our work addresses general stochastic control problems with nonlinear dynamics and compact action spaces, and develops a rigorous $L^2$-error analysis aligned with residual loss minimization. We prove exponential convergence under classical policy iteration, offering a quantitative decomposition of total error that accounts for both approximation and policy mismatch.

## 3 INFINITE-HORIZON STOCHASTIC OPTIMAL CONTROL

Let $(W_t)_{t \geq 0}$ be a $d$-dimensional Brownian motion on a filtered probability space $(\Omega, \mathcal{F}, \mathcal{F}_t, \mathbb{P})$. A bounded and measurable control process $a_t \in A \subset \mathbb{R}^m$ drives the controlled diffusion

$$dX_t = b(X_t, a_t)\, \mathrm{d}t + \sigma\, \mathrm{d}W_t, \qquad X_0 = x \in \mathbb{R}^d, \tag{1}$$

where $\sigma$ is a constant matrix. The goal is to maximize the infinite-horizon discounted cost

$$J(x, a) = \mathbb{E}_x \left[ \int_0^\infty e^{-\lambda s} L(X_s, a_s)\, \mathrm{d}s \right], \qquad \lambda > 0. \tag{2}$$

With the value function defined as $V(x) := \sup_a J(x,a)$, it is known from literature Tran (2021); Evans (2022) that $V \in \text{Lip}(\mathbb{R}^d)$ is a unique viscosity solution to

$$\lambda V - \frac{1}{2}\text{tr}(\sigma\sigma^\top D_{xx}^2 V) - \sup_{a \in A}\{b \cdot \nabla_x V + L\} = 0, \tag{3}$$

under some regularity assumptions on $b, L$. Then the optimal control is given by

$$a^*(x) := \arg\max_{a \in A}\{b(x,a) \cdot \nabla_x V(x) + L(x,a)\},$$

where measurable selection guarantees the measurability of the control.

In the remainder of this section, we formalize the mathematical setting, assumptions, and notation used throughout the paper.

**Notations** Let us begin by introducing the notations used throughout the paper. For $x \in \mathbb{R}^d$, we write $|x|$ for the Euclidean norm. Given a function $f : \Omega \to \mathbb{R}^n$, we denote its standard $L^p$ norm by

$$\|f\|_p := \left(\int_\Omega |f|^p \, dx\right)^{1/p}, \quad \text{for} \quad p \in [1, \infty].$$

The Hessian of $f$ is denoted $D_{xx}^2 f$. We say $f \in H^1(\mathbb{R}^d) = W^{1,2}(\mathbb{R}^d)$ if $\int_\Omega |f|^2 + |\nabla_x f|^2 < \infty$. For $g : \mathbb{R}^d \to \mathbb{R}^d$, $\text{div}_x g := \sum_{i=1}^d \partial_{x_i} g_i$ where $g_i$ denotes the $i$th component of $g$.

**Assumption 1.** *We impose the following assumptions throughout the paper.*

*(A1) Control set $A \subset \mathbb{R}^d$ is compact and convex.*

*(A2) $b : \mathbb{R}^d \times A \to \mathbb{R}$ are continuously differentiable and Lipschitz continuous. In addition, $b(x,a)$ satisfies:*

- *$\lambda > B/2$ where*

$$B := \sup_{a \in A}(\|b(\cdot, a)\|_\infty + \|\text{div}_x b(\cdot, a)\|_\infty) < \infty,$$

- *the Jacobian $\partial_a b(x,a)$ is uniformly bounded,*

- *there exists a constant $\tilde{B} > 0$ satisfying*

$$|\partial_a b(x,a)| + \frac{|\partial_a b(x,a) - \partial_a b(x,a')|}{|a - a'|} \leq \tilde{B}.$$

*(A3) $L(x,a) \geq 0$ is uniformly $L_a$-Lipschitz continuous and $\mu_a$-strongly convex in $a$. Furthermore, there exist constants $R > 0$, $\beta > d + 2$, and $C > 0$ such that*

$$\sup_{a \in A} L(x,a) \leq \frac{C}{(1 + |x|)^\beta}, \qquad \text{for all } |x| \geq R.$$

*(A4) $\sigma\sigma^\top$ is uniformly elliptic with eigenvalues bounded between $0 < \nu \leq \Lambda$.*

Assumption (A3) ensures that the running cost $L(x,a)$ decays sufficiently fast at infinity, while (A4) guarantees nondegenerate diffusion. These together imply that the value function $V(x) = \sup_a \mathbb{E}_x[\int_0^\infty e^{-\lambda t} L(X_t, a_t) \, dt]$ is integrable over $\mathbb{R}^d$, i.e., $V \in L^2(\mathbb{R}^d)$, via standard estimates on stochastic processes with confining cost structure.

## 4 HOWARD'S POLICY IMPROVEMENT ALGORITHM

---
**Algorithm 1** Policy Improvement

---
1: **Input:** initial Markov policy $a_0(\cdot)$.
2: **for** $n = 0, 1, 2, \ldots$ **do**
3:     *Policy evaluation:* solve for $v_n$ on $\mathbb{R}^d$

$$\lambda v_n - \tfrac{1}{2} \operatorname{tr}(\sigma \sigma^\top D_{xx}^2 v_n) - b(\cdot, a_n) \cdot \nabla_x v_n = L(\cdot, a_n).$$

4:     *Policy improvement:*

$$a_{n+1}(x) = \arg\max_{a \in A} \{L(x, a) + b(x, a) \cdot \nabla_x v_n(x)\}.$$

5:     **if** $\|v_{n+1} - v_n\|_\infty < \varepsilon$ **then stop**.
6: **end for**

---

Howard's policy iteration alternates between evaluating the cost of a fixed policy and improving it by acting greedily with respect to the current value function. This structure naturally aligns with model-based reinforcement learning methods and enables interpretable control synthesis in continuous domains. Unlike value iteration, which updates the value function directly via a fixed-point operator, policy iteration produces stable value approximations by solving a linear PDE at each step.

The key advantage of this method lies in the decoupling of policy evaluation and improvement: the former reduces to solving a linear PDE, and the latter often admits a closed-form optimizer when $L(x, a) + b(x, a) \cdot \nabla_x V(x)$ is convex in $a$. This makes PI especially appealing for structured control problems where the policy improvement step can be implemented efficiently.

However, the main computational bottleneck in Howard's method lies in solving the high-dimensional linear PDE in each policy evaluation step. Traditional finite-difference or finite-element schemes scale poorly in high dimensions. In the next section, we propose to overcome this limitation using physics-informed neural networks (PINNs), which serve as flexible, mesh-free solvers capable of approximating solutions in high-dimensional domains.

## 5 PHYSICS-INFORMED HOWARD POLICY ITERATION

We propose a PINN-based variant of Howard's policy iteration, where the value function at each iteration is approximated by a neural network trained to minimize the PDE residual at sampled collocation points. Let $\{x_i\}_{i=1}^N \subset \Omega$ be a set of $N$ collocation points sampled from the domain.[1] Given a fixed policy $a_n(x)$, we approximate the corresponding value function $v_n(x)$ by a neural network $v_n(x; \theta)$ and define the residual of the linear PDE:

$$\mathcal{L}(\theta) := \frac{1}{N} \sum_{i=1}^N \big| \lambda v_n(x_i; \theta) - \tfrac{1}{2} \operatorname{tr}(\sigma \sigma^\top D_{xx}^2 v_n(x_i; \theta)) \tag{4}$$
$$- b(x_i, a_n(x_i)) \cdot \nabla_x v_n(x_i; \theta) - L(x_i, a_n(x_i)) \big|^2$$

Before analyzing the convergence of our proposed method, we first establish a stability result for the linear PDE solved at each policy evaluation step. Specifically, for a fixed measurable policy $a_n \colon \mathbb{R}^d \to A$, the value function $v_n$ satisfies a linear elliptic PDE of the form:

$$\lambda v_n - \tfrac{1}{2} \operatorname{tr}(\sigma \sigma^\top D_{xx}^2 v_n) - b_n \cdot \nabla_x v_n = k,$$

where $b_n(x) := b(x, a_n(x))$ and $k(x) \in L^2(\mathbb{R}^d)$ is a given forcing term. The following proposition shows that under mild assumptions, this equation admits a unique weak solution $v_n \in H^1(\mathbb{R}^d)$, with an energy estimate that is uniform in the choice of the measurable policy $a_n$. This result plays a key role in subsequent error analysis.

---

[1]Although the theoretical analysis is conducted over $\mathbb{R}^d$, we assume a bounded domain $\Omega \subset \mathbb{R}^d$ in practice since only a finite number of collocation points are sampled.

---

**Algorithm 2** Physics-Informed Neural Network Policy Iteration (PINN-PI)

---

1: **Input:** Initial policy $a_0(\cdot)$, number of collocation points $N$, domain $\Omega$, initial network parameters $\theta_0$
2: **for** $n = 0, 1, 2, \ldots$ **do**
3:     **Collocation sampling:** Sample $\{x_i\}_{i=1}^N \subset \Omega$
4:     **Policy evaluation:** Train neural network $v_n(x; \theta)$ by minimizing the residual loss defined in equation 4
5:     **Policy improvement:**

$$a_{n+1}(x) := \arg\max_{a \in A} \{L(x, a) + b(x, a) \cdot \nabla_x v_n(x; \theta_n)\}$$

6:     **If** stopping criterion met (e.g. $\|a_{n+1} - a_n\|_\infty < \varepsilon$) **then stop**
7: **end for**

---

**Proposition 1** ($L^2$ estimate with a measurable policy $a_n$). *Suppose Assumption 1 holds. Let $a_n \colon \mathbb{R}^d \to A$ be any measurable policy and set $b_n(x) := b(x, a_n(x))$. For $k \in L^2(\mathbb{R}^d)$ consider the PDE*

$$\lambda v_n - \tfrac{1}{2}\operatorname{tr}(\sigma\sigma^\top D_{xx}^2 v_n) - b_n \cdot \nabla_x v_n = k \quad in \quad \mathbb{R}^d.$$

*Then there is a unique weak solution $v_n \in H^1$ satisfying*

$$\left(\lambda - \tfrac{1}{2}B\right)\|v\|_2^2 + \frac{\nu}{2}\|\nabla_x v\|_2^2 \le (k, v_n),$$

*where $(f, g) := \int_{\mathbb{R}^d} fg\, \mathrm{d}x$. Therefore,*

$$\|v\|_2 \le C_\lambda \|\tilde{r}\|_2, \quad \|\nabla_x v\|_2 \le C_\lambda \|k\|_2,$$

*where $C_\lambda = \max\{\frac{1}{\lambda - \frac{1}{2}B}, \sqrt{\frac{1}{\nu(\lambda - \frac{1}{2}B)}}\}$.*

For the sake of completeness, the proof is provided in Appendiex A.1 While the proposition above ensures the stability of each policy evaluation step in the $L^2$ sense, it does not by itself guarantee that the updated policy improves over iterations. To analyze the overall convergence behavior of policy iteration, it is crucial to understand how the quality of the value function approximation, particularly its gradient, affects the resulting policy.

To this end, the next proposition shows that the policy improvement map is Lipschitz continuous with respect to the value gradient. This result allows us to quantify how errors in the value approximation propagate to the policy error in a stable manner, which is a key ingredient in establishing exponential convergence.

**Proposition 2** (Policy error controlled by value–gradient error). *Assume (A1)–(A4) and let $|z|, |z'| \le M$ for some $M$ such that $\mu_a > M\tilde{B}$. Fix $x \in \mathbb{R}^d$ and define the selector*

$$a^*(x, z) := \arg\max_{a \in A}\Big\{L(x, a) + b(x, a) \cdot z\Big\}, \qquad z \in \mathbb{R}^d.$$

*Then $a^*$ is globally Lipschitz in $z$ with constant $\theta > 0$:*

$$|a^*(x, z) - a^*(x, z')| \le \theta|z - z'| \quad for \quad z, z' \in \mathbb{R}^d.$$

The proof is presented in Appendix A.2.

In Kerimkulov et al. (2020), the pointwise exponential convergence has been established, which yields that

$$0 \le v_n(x) - V(x) \le C\eta^n, \qquad \forall x \in \mathbb{R}^d,$$

for some $\eta \in (0, 1)$. However, in the framework of PINNs, $L^2$ is more suitable so, we now establish the exponential convergence property of $v_n$ to $V$ in $L^2$.

Throughout this section, we use

$$C_R := \theta(L_a + \tilde{B} \max_{p \in \{2, \infty\}} \{\|\nabla_x V\|_p, \|\nabla_x v_n\|_p\}),$$

where $V$ is a unique solution to equation 3 and $\{v_n\}_{n \geq 0}$ is generatved via Algorithm 1. Here, by classical theory of elliptic PDEs Evans (2022) and the Lipschitz continuity of $V$, this $C_R$ is finite. For brevity, let us define

$$\mathcal{T}[v, a] := \lambda v - \tfrac{1}{2}\operatorname{tr}(\sigma\sigma^\top D_{xx}^2 v) - b(\cdot, a) \cdot \nabla_x v - L(\cdot, a).$$

**Theorem 1** (Global exponential convergence of Howard–PI). *Let Assumption 1 hold and $V$ be a unique viscosity solution to equation 3 with continuous gradient. If $\tilde{\kappa} := \sqrt{\dfrac{C_R^2}{\nu(\lambda - \frac{1}{2}B)}} \in (0, 1)$, then we have*

$$\|v_n - V\|_2 \leq C\tilde{\kappa}^n. \qquad x \in \mathbb{R}^d,$$

*where $\{(v_n, a_n)\}_{n \geq 0}$ be produced by Algorithm 1, and $C$ is a problem dependent constant.*

We introduce a lemma that links the value functions and their gradients.

**Lemma 1.** *With $C_R$ defined above, we have that*

$$\|v_n - v_m\|_2 \leq \tilde{C}_\lambda \|\nabla_x v_{n-1} - \nabla_x v_{m-1}\|_2,$$

*and*

$$\|\nabla_x v_n - \nabla_x v_m\|_2 \leq \tilde{C}_\lambda \|\nabla_x v_{n-1} - \nabla_x v_{m-1}\|_2,$$

*where $\tilde{C}_\lambda = \max\{\dfrac{C_R}{\lambda - \frac{1}{2}B}, \sqrt{\dfrac{C_R^2}{\nu(\lambda - \frac{1}{2}B)}}\}$.*

The proof of this lemma is provided in Appendix A.3

*Proof of Theorem 1.* Recalling

$$a^*(x) := \arg\max_{a \in A}\{b(x, a) \cdot \nabla_x V(x) + L(x, a)\},$$

$V \in \operatorname{Lip}(\mathbb{R}^d) \cap L^2(\mathbb{R}^d)$ is a unique viscosity solution to

$$\lambda V - \tfrac{1}{2}\operatorname{Tr}(\sigma\sigma^\top D_{xx}^2 V) - b(\cdot, a^*) \cdot \nabla_x V - L(\cdot, a^*) = 0.$$

Subtracting from $\mathcal{T}[v_n, a_n]$, we achieve

$$\lambda e_n - \tfrac{1}{2}\operatorname{Tr}(\sigma\sigma^\top D_{xx}^2 e_n) - b(\cdot, a_n) \cdot \nabla_x e_n = R_n,$$

where $e_n := v_n - V$ and $R_n := [b(\cdot, a_n) - b(\cdot, a^*)] \cdot \nabla_x V + [L(\cdot, a_n) - L(\cdot, a^*)]$.

By applying Lemma 1 with $v_n$ and $v_m = V$, we deduce that

$$\|\nabla_x e_n\|_2 \leq \sqrt{\frac{C_R^2}{\nu(\lambda - \frac{1}{2}B)}}\|\nabla_x e_{n-1}\|_2,$$

and hence,

$$\|\nabla_x e_n\|_2 \leq \kappa^n \|\nabla_x e_0\|_2.$$

Now from

$$\|e_n\|_2 \leq \frac{C_R}{\lambda - \frac{1}{2}B}\|\nabla_x e_{n-1}\|_2,$$

we conclude that

$$\|e_n\|_2 \leq C\kappa^n.$$

for some global constant $C > 0$. $\qquad\qquad\square$

Finally, combining Theorem 1 and Proposition 3 introduced below, we arrive at a quantitative bound on the total approximation error of the PINN-based policy iteration method. Specifically, with $\{\tilde{v}_n\}_{n \geq 0}$ be generated via Algorithm 2, we define the three error components

$$\delta_n := \tilde{v}_n - \hat{v}_n, \quad \varepsilon_n := \hat{v}_n - v_n, \quad \epsilon := v_n - V, \tag{5}$$

so that $\tilde{e}_n = \delta_n + \varepsilon_n + e_n$, where $e_n := v_n - V$ is the ideal policy-iteration error. There exists a (user-chosen) tolerance $p_n > 0$ such that

$$\|r_n\|_2 \leq p_n, \qquad n = 0, 1, \cdots, \tag{6}$$

where

$$r_n := \lambda \tilde{v}_n - \tfrac{1}{2} \operatorname{tr}(\sigma \sigma^\top D_{xx}^2 \tilde{v}_n) - b(\cdot \tilde{a}_n) \cdot \nabla_x \tilde{v}_n - L(\cdot, \tilde{a}_n). \tag{7}$$

The distinction between $\hat{v}_n$ and $\tilde{v}_n$ is essential for understanding the approximation error introduced by the PINN surrogate, $\tilde{v}_n$. Specifically, $\hat{v}_n$ denotes the exact solution to the linear PDE associated with the frozen policy $\tilde{a}_n$ derived from $\tilde{v}_{n-1}$, which means that $\mathcal{T}[\hat{v}_n, \tilde{a}_n] = 0$. In contrast, $\tilde{v}_n$ is the neural approximation to $\hat{v}_n$ obtained by minimizing the residual loss equation 4 at a finite set of collocation points. Therefore, $\delta_n = \tilde{v}_n - \hat{v}_n$ captures the discrepancy due to numerical training, discretization, and model capacity limitations of the PINN. Importantly, $\delta_n$ is fully controlled by the optimization procedure and serves as the primary source of empirical error in our framework.

**Proposition 3** (Policy–mismatch recursion)**.** *Let* $\{(\tilde{v}_n, \tilde{a}_n)\}_{n \geq 0}$ *be generated via Algorithm 2, and* $\kappa := \tilde{C}_\lambda \in (0, 1)$ *with* $\lambda$ *sufficiently large. Then, under the same assumption as in Theorem 1, we have that*

$$\|\delta_n\|_2 + \|\varepsilon_n\|_2 \leq C(p + \kappa^n). \tag{8}$$

*for some problem dependent constant* $C > 0$ *where* $p = \sup_n p_n$.

*Proof.* To estimate $\delta_n$, we recall that $\mathcal{T}[\tilde{v}_n, \tilde{a}_n] = r_n$ with $r_n$ defined in equation 4 and $\mathcal{T}[\hat{v}_n, \tilde{a}_n] = 0$. Subtracting two, with $C_\lambda$ from Proposition 1, we have

$$\|\delta_n\|_2 \leq C_\lambda p_n \leq C_\lambda p.$$

We now estimate $\varepsilon = \hat{v}_n - v_n$. Noting that $\hat{v}_n$ and $v_n$ satisfy $\mathcal{T}[\hat{v}_n, \tilde{a}_n] = 0$ and $\mathcal{T}[v_n, a_n]=0$, we invoke Lemma 1 with $\hat{v}_n$ and $v_n$. Hence,

$$\|\varepsilon\|_2 = \|\hat{v}_n - v_n\|_2 \leq C_\lambda \|\nabla_x \tilde{v}_{n-1} - \nabla_x v_{n-1}\|,$$

since $\tilde{a}_n$ is induced by $\tilde{v}_{n-1}$.

Applying Lemma 1 once again with $\tilde{v}_{n-1}$ and $v_{n-1}$, we have

$$\|\nabla_x \tilde{v}_{n-1} - \nabla_x v_{n-1}\|_2 \leq \|\nabla_x \tilde{v}_{n-1} - \nabla_x \hat{v}_{n-1}\|_2 + \|\nabla_x \hat{v}_{n-1} - \nabla_x v_{n-1}\|_2$$
$$\leq C_\lambda p_{n-1} + \tilde{C}_\lambda \|\nabla_x \tilde{v}_{n-2} - \nabla_x v_{n-2}\|$$

Denoting $g_n := \|\nabla_x \tilde{v}_n - \nabla_x v_n\|_2$, we have that

$$g_{n-1} \leq C_\lambda p_{n-1} + \tilde{C}_\lambda g_{n-2},$$

which leads to

$$g_{n-1} \leq (\tilde{C}_\lambda)^{n-1} g_0 + C_\lambda p \sum_{i=0}^{n-1} (\tilde{C}_\lambda)^i \leq g_0 \kappa^{n-1} + \frac{C_\lambda p}{1 - \kappa}.$$

since $\kappa = \tilde{C}_\lambda \in (0, 1)$. Therefore,

$$\|\nabla_x \tilde{v}_{n-1} - \nabla_x v_{n-1}\|_2 \leq \kappa^{n-1} \|\nabla_x \tilde{v}_0 - \nabla_x v_0\|_2 + \frac{C_\lambda p}{1 - \kappa},$$

and thereby,

$$\|\varepsilon_n\|_2 \leq C(\kappa^n + p),$$

for some $C > 0$. $\qquad\square$

Theorem 1 establishes exponential convergence of Howard's method under exact policy evaluation. In practice, however, our PINN-based framework introduces approximation errors due to finite training, neural network capacity, and collocation sampling. These errors manifest as discrepancies between the neural surrogate $\tilde{v}_n$ and the exact PDE solution $v_n$ at each iteration.

To rigorously quantify the cumulative effect of these approximations, we now derive a global $L^2$ error bound that separates the ideal contraction behavior from the training-induced deviations. This result justifies the robustness of our approach and provides guidance on the choice of residual tolerance during training.

**Theorem 2** (Global $L^2$ error bound). *Under the same assumption in Proposition 3 and $\tilde{\kappa} \in (0, 1)$, we have that*

$$\|\tilde{v}_n - V\|_2 \leq C(p + \kappa^n + \tilde{\kappa}^n), \tag{9}$$

*where $\tilde{\kappa}$ is from Theorem 1.*

*Proof.* The proof immediately follows from Theorem 1 and Proposition 3 after decomposition

$$\tilde{e}_n = \delta_n + \varepsilon_n + (v_n - V).$$

$\square$

This result confirms that the overall error between the neural approximation $\tilde{v}_n$ and the optimal value function $V$ can be decomposed into a controllable training error and an exponentially decaying ideal iteration error. In particular, so long as the residual tolerance $p_n$ remains uniformly bounded, the cumulative error remains stable across iterations. This theoretical guarantee forms the basis for choosing the training accuracy of the PINN at each step in practice.

In the next section, we empirically validate these theoretical insights on a range of benchmark control problems, demonstrating both the convergence behavior and the accuracy of the resulting policies.

## 6 EXPERIMENTS

We empirically validate our proposed PINN-based policy iteration (PINN-PI) framework, which implements a physics-informed variant of Howard's policy iteration scheme for stochastic optimal control. Our experiments span both linear-quadratic and nonlinear benchmark systems with stochastic dynamics and compact action spaces. Through these experiments, we aim to demonstrate: (1) scalability and stability of PINN-PI in high-dimensional settings, including monotonicity of value functions Howard (1960); Kerimkulov et al. (2020), (2) its advantage over model-free baselines such as SAC (3), and its robustness.

### 6.1 LINEAR-QUADRATIC REGULATOR (LQR) WITH COMPACT ACTION SPACE

We begin with stochastic LQR problems in dimensions $d = m \in \{5, 10, 20\}$ under a compact control set. The compactness assumption breaks the standard Riccati structure and eliminates closed-form solutions, yet the true value function remains nearly quadratic, making this setting a useful benchmark for evaluating learning performance. To evaluate the learned policies, we report rewards averaged over 30 trajectories starting from the zero initial state.

As a model-free comparison, we train Soft Actor-Critic (SAC) Haarnoja et al. (2018) on the same problem, using identical initializations and noise realizations. Unlike PINN-PI, SAC must discover both dynamics and cost structure purely from rollouts.

Figure 1 compares our method (PINN-PI) with Soft Actor-Critic (SAC) in 5D and 10D LQR settings. As demonstrated, PINN-PI consistently achieves higher reward and smoother convergence, while SAC struggles to generalize in high dimensions due to sample inefficiency. The result with $(d, m) = (20, 20)$ is provided in Appendix B together with experimtnal details.

### 6.2 NONLINEAR BENCHMARKS WITH STOCHASTIC DYNAMICS

To evaluate performance in more realistic and nonlinear scenarios, we consider two widely used benchmark environments Brockman et al. (2016): the stochastic inverted pendulum and cartpole. Both systems are modeled as stochastic control-affine dynamics with additive Brownian noise.

Figure 2 shows the evolution of performance over training, evaluated from 10 trajectory samples with the same random initial states. PINN-PI consistently stabilizes the system faster and achieves higher reward than SAC, while strictly enforcing control constraints. Notably, in high-noise regimes, SAC exhibits oscillatory behavior due to imperfect reward shaping, whereas PINN-PI produces smoother, more stable trajectories by leveraging model information and HJB structure. The monotonicity property of policy iteration is also confirmed in both tasks.

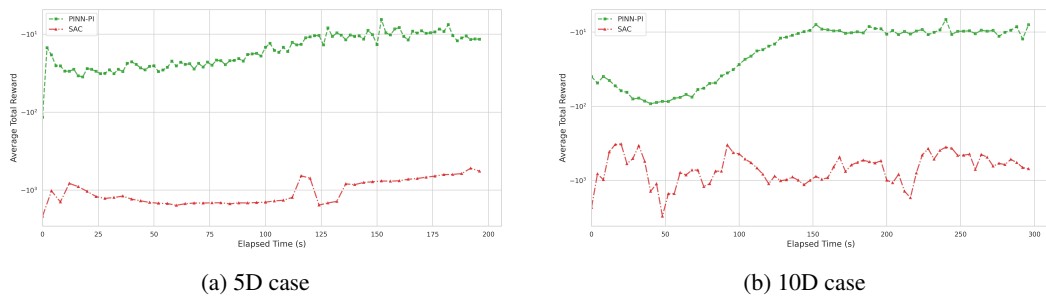

(a) 5D case          (b) 10D case

Figure 1: Comparison between PINN-PI (ours) and SAC in learning stochastic LQR problems with compact control sets. PINN-PI exhibits monotonic improvement of the value function, while SAC shows less stable learning behavior.

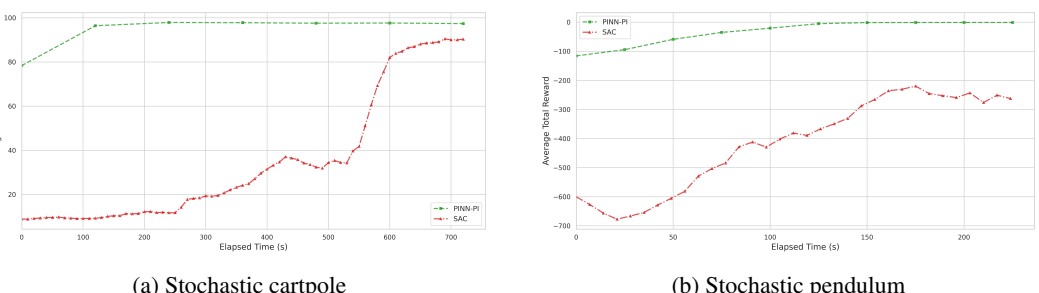

(a) Stochastic cartpole          (b) Stochastic pendulum

Figure 2: Comparison between PINN-PI and SAC. In both cases, PINN-PI exhibits a monotonic increase in the value function.

# 7 DISCUSSION

Our method extends the deterministic and affine-in-control setting of Meng et al. (2024) by establishing an $L^2$-based convergence theory for general stochastic control problems with nonlinear dynamics and compact action spaces. A key feature of our method is that the total approximation error across iterations remains uniformly bounded, enabling systematic monitoring of policy quality through $L^2$ energy estimates.

While our approach requires full model knowledge (drift and diffusion), once the HJB equation is solved through our PINN-based framework, the resulting value function immediately yields the optimal policy for any new initial state. In this sense, our method leverages model information to produce a reusable global solution, in contrast to model-free methods that re-train for each task instance. This property highlights the scalability of our framework: solving a high-dimensional HJB not only establishes theoretical guarantees but also provides a direct mechanism for policy recovery across state space.

Several practical directions remain open, including reducing the cost of policy improvement for nonlinear dynamics, developing dimension-adaptive sampling schemes, and extending the framework to settings with partially unknown dynamics. Nonetheless, the ability to recover optimal policies universally from a trained value function underscores the broader impact of combining PDE theory with physics-informed learning in high-dimensional control.

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
