## REPRODUCIBILITY STATEMENT

We have taken several steps to ensure the reproducibility of our results. All theoretical claims are stated with clear assumptions in Section 3 and rigorously proved in Appendix (A.1– A.3). The complete description of our proposed PINN-based policy iteration algorithm is provided in Algorithm 2 (Section 5), together with detailed error analysis and convergence guarantees (Theorems 1–2). Experimental setups, including hyperparameters, network architectures, and collocation sampling strategies, are fully described in Section 6 and Appendix B. To further facilitate reproducibility, we will release anonymous source code and scripts for reproducing all experiments (LQR, pendulum, cartpole) as supplementary material.

## A  APPENDIX

### A.1  PROOF OF PROPOSITION 1

Since $a_n$ takes values in $A$ and $b$ is uniformly bounded on $\mathbb{R}^d \times A$, we have $\|b_n\|_\infty \leq B$ and $\|\mathrm{div}_x\, b_n\|_\infty \leq B$. For $u, \varphi \in H^1(\mathbb{R}^d)$, we set

$$a(u, \varphi) := \tfrac{1}{2} \int_{\mathbb{R}^d} \sigma\sigma^\top \nabla_x u \cdot \nabla_x \varphi + \lambda \int_{\mathbb{R}^d} u\varphi - \int_{\mathbb{R}^d} b_n \cdot \nabla_x u\varphi.$$

The right-hand side is $\ell(\varphi) := \int k\varphi$. We now see the boundedness of the coercivity of $a$ as

$$|a(u, \varphi)|$$
$$\leq \tfrac{\Lambda}{2}\|\nabla_x u\|_2 \|\nabla_x \varphi\|_2 + \lambda\|u\|_2\|\varphi\|_2 + B\|\nabla_x u\|_2\|\varphi\|_2$$
$$\leq C_1 \|u\|_{H^1}\|\varphi\|_{H^1}.$$

and

$$a(u, u) = \tfrac{1}{2}\int \nabla_x u^\top \sigma\sigma^\top \nabla_x u + \lambda\|u\|_2^2 - \tfrac{1}{2}\int (\mathrm{div}\, b_n)u^2$$
$$\geq \tfrac{\nu}{2}\|\nabla_x u\|_2^2 + (\lambda - \tfrac{1}{2}B)\|u\|_2^2.$$

Since $\lambda > \tfrac{1}{2}B$, the form is coercive.

The functional $\ell$ is continuous on $H^1$. By the Lax–Milgram theorem a unique $v_n \in H^1(\mathbb{R}^d)$ solves $a(v_n, \varphi) = \ell(\varphi)$ for all $\varphi$.

To finish the energy estimate, we test inequality with $\varphi = v_n$, which leads to

$$\lambda\|v_n\|_2^2 + \tfrac{\nu}{2}\|\nabla_x v_n\|_2^2 \leq \tfrac{1}{2}B\|v_n\|_2^2 + (k, v_n).$$

Continuing from above, rearranging the inequality gives:

$$\left(\lambda - \tfrac{1}{2}B\right)\|v_n\|_2^2 + \tfrac{\nu}{2}\|\nabla_x v_n\|_2^2 \leq (k, v_n),$$

where $(f, g) := \int_{\mathbb{R}^d} f(x)g(x)\,\mathrm{d}x$.

We now estimate the right-hand side using Cauchy–Schwarz and Young's inequality. For any $\varepsilon > 0$, we have:

$$(k, v_n) \leq \|k\|_2\|v_n\|_2 \leq \varepsilon\|v_n\|_2^2 + \frac{1}{4\varepsilon}\|k\|_2^2.$$

Substituting this into the inequality, we obtain:

$$\left(\lambda - \tfrac{1}{2}B - \varepsilon\right)\|v_n\|_2^2 + \tfrac{\nu}{2}\|\nabla_x v_n\|_2^2 \leq \frac{1}{4\varepsilon}\|k\|_2^2.$$

Taking $\varepsilon = \tfrac{1}{2}(\lambda - \tfrac{1}{2}B)$, which is valid because $\lambda > \tfrac{1}{2}B$, yields:

$$\|v_n\|_2^2 \leq \frac{1}{(\lambda - \tfrac{1}{2}B)^2}\|k\|_2^2, \quad \|\nabla_x v_n\|_2^2 \leq \frac{1}{\nu(\lambda - \tfrac{1}{2}B)}\|k\|_2^2.$$

## A.2 PROOF OF PROPOSITION 2

Let $z, z' \in \mathbb{R}^d$ and denote
$$a := a^*(x, z), \qquad a' := a^*(x, z').$$

By the definition of $a^*$ as a maximizer over a convex set $A$, and the strong convexity of the objective function, the maximizers $a, a'$ are unique and continuous.

The necessary condition for optimality (first-order variational inequality) yields:
$$\langle \nabla_a L(x, a) + \partial_a b(x, a)^\top z, a' - a \rangle \geq 0, \tag{10}$$
$$\langle \nabla_a L(x, a') + \partial_a b(x, a')^\top z', a - a' \rangle \geq 0. \tag{11}$$

Adding equation 10 and equation 11 gives:
$$\langle \nabla_a L(x, a) - \nabla_a L(x, a'), a' - a \rangle$$
$$+ \langle [\partial_a b(x, a) - \partial_a b(x, a')]^\top z, a' - a \rangle$$
$$+ \langle \partial_a b(x, a')^\top (z - z'), a' - a \rangle \geq 0.$$

Now use the $\mu_a$-strong convexity of $L$ in $a$:
$$\langle \nabla_a L(x, a) - \nabla_a L(x, a'), a - a' \rangle \geq \mu_a |a - a|^2.$$

Therefore, we obtain:
$$\mu_a |a - a'|^2 \leq |\langle [\partial_a b(x, a) - \partial_a b(x, a')]^\top z, a - a' \rangle|$$
$$+ |\langle \partial_a b(x, a')^\top (z - z'), a - a' \rangle|.$$

Since $\partial_a b$ is $\tilde{B}$-Lipschitz in $a$, the first term becomes
$$|\langle [\partial_a b(x, a) - \partial_a b(x, a')]^\top z, a - a' \rangle| \leq \tilde{B} |z| |a - a'|^2.$$

The second term is handled via
$$|\langle \partial_a b(x, a')^\top (z - z'), a - a' \rangle| \leq \tilde{B} |z - z'| |a - a'|.$$

Combine the bounds:
$$\mu_a |a - a'|^2 \leq \tilde{B} |z| |a - a'|^2 + \tilde{B} |z - z'| |a - a'|.$$

Now, subtract $\tilde{B} |z| |a - a'|^2$ from both sides:
$$(\mu_a - \tilde{B} |z|) |a - a'|^2 \leq \tilde{B} |z - z'| |a - a'|.$$

Since $\mu_a > \tilde{B} |z|$, we can divide both sides by $|a - a'|$:
$$|a - a'| \leq \frac{\tilde{B}}{\mu_a - \tilde{B} |z|} |z - z'|.$$

Therefore,
$$|a - a'| \leq \theta |z - z'|,$$
for some $\theta > 0$.

## A.3 PROOF OF LEMMA 1

Recall that $\mathcal{T}[v_n, a_n] = \mathcal{T}[v_m, a_m] = 0$ and subtract two equations to achieve
$$\lambda e - \tfrac{1}{2} \operatorname{Tr}(\sigma \sigma^\top D_{xx}^2 e) - b(\cdot, a_n) \cdot \nabla_x e = R,$$
where $e := v_n - v_m$ and
$$R := [b(\cdot, a_n) - b(\cdot, a_m)] \cdot \nabla_x v_m + [L(\cdot, a_n) - L(\cdot, a_m)].$$

We now test with respect to $e$ and proceed with the $L^2$ coercivity argument of Proposition 1 yields

$$(\lambda - \tfrac{1}{2}B)\|e\|_2^2 + \tfrac{\nu}{2}\|\nabla_x e\|_2^2 \le (R, e), \tag{12}$$

where $(f, g) := \int_{\mathbb{R}^d} fg \, \mathrm{d}x$. Now the right-hand side of the inequality is estimated as

$$\|R\|_2 \le \tilde{B}\|a_n - a_m\|_2\|\nabla_x v_m\|_2 + L_a\|a_n - a_m\|_2$$
$$\le \underbrace{\theta(\tilde{B}\sup_m \|v_m\|_2 + L_a)}_{\le C_R}\|\nabla_x v_{n-1} - \nabla_x v_{m-1}\|_2$$

where $\theta$ is from Proposition 2. Hence, we have that

$$\|R\|_2 \le C_R\|\nabla_x v_{n-1} - \nabla_x v_{m-1}\|_2,$$

and therefore,

$$(R, e) \le C_R\|\nabla_x v_{n-1} - \nabla_x v_{m-1}\|_2\|e\|_2 \tag{13}$$

by Cauchy–Schwarz inequality.

Applying the Young's inequality $ab \le \tfrac{\varepsilon}{2}a^2 + \tfrac{1}{2\varepsilon}b^2$ with $\varepsilon = \frac{C_R}{2(\lambda - \tfrac{1}{2}B)}$, $a = \nabla_x v_{n-1} - \nabla_x v_{m-1}$, and $b = e$ in equation 13, we deduce that

$$\tfrac{\nu}{2}\|\nabla_x e\|_2^2 \le \frac{C_R^2}{2(\lambda - \tfrac{1}{2}B)}\|\nabla_x v_{n-1} - \nabla_x v_{m-1}\|_2^2,$$

and hence,

$$\|\nabla_x e\|_2 \le \sqrt{\frac{C_R^2}{\nu(\lambda - \tfrac{1}{2}B)}}\|\nabla_x v_{n-1} - \nabla_x v_{m-1}\|_2.$$

On the other hand, observing $\|e\|_2$ explicitly, we have

$$(\lambda - \tfrac{1}{2}B)\|e\|_2^2 \le \|R\|_2\|e\|_2$$
$$\le C_R\|\nabla_x v_{n-1} - \nabla_x v_{m-1}\|_2\|e\|_2.$$

Canceling $\|e\|_2$, we get

$$\|e\|_2 \le \frac{C_R}{\lambda - \tfrac{1}{2}B}\|\nabla_x v_{n-1} - \nabla_x v_{m-1}\|_2.$$

## B  EXPERIMENTAL DETAILS AND ADDITIONAL RESULT

For all experiments, we set the discount factor $\lambda = 0.5$ and the temporal discretization step to $dt = 0.05$, which yields $\exp(-\lambda dt) \approx 0.99$.

**Stochastic linear-quadratic regulator.**  For the stochastic LQR experiments, we considered systems of dimension $(d, m) = (5, 5), (10, 10), (20, 20)$ with the following dynamics:

$$\mathrm{d}X_t = (AX_t + Bu_t)\,\mathrm{d}t + \sigma\,\mathrm{d}W_t,$$

where $\sigma = 0.1 \cdot I_d$ and $u_t \in A := \{u \in \mathbb{R}^m \mid \|u\|_\infty \le \mathbf{u}\}$ with $\mathbf{u} = 1$. For all cases, the reward function is set to

$$L(x, u) = -x^\top Q x - u^\top R u,$$

where $Q = 5I_d \succ 0$ and $R = I_d \succ 0$.

For the case $(d, m) = (5, 5)$, the system dynamics $A$ and $B$ are chosen as:

$$A = \begin{bmatrix} 0.4455 & 0.2996 & 0.4497 & 0.2813 & 0.3114 \\ 0.4317 & 0.2885 & 0.3214 & 0.2523 & 0.4193 \\ 0.1975 & 0.4295 & 0.3014 & 0.4225 & 0.2282 \\ 0.4361 & 0.4716 & 0.3812 & 0.3010 & 0.0037 \\ 0.2247 & 0.0629 & 0.3350 & 0.0424 & 0.1836 \end{bmatrix},$$

$$B = \begin{bmatrix} 0.0681 & 0.0544 & 0.0467 & 0.0152 & 0.0787 \\ 0.0970 & 0.0081 & 0.0145 & 0.0034 & 0.0984 \\ 0.0358 & 0.0833 & 0.0324 & 0.0839 & 0.0012 \\ 0.0116 & 0.0280 & 0.0056 & 0.0092 & 0.0432 \\ 0.0047 & 0.0848 & 0.0718 & 0.0977 & 0.0556 \end{bmatrix}.$$

For the case $(d, m) = (10, 10)$, the system matrices $A$ and $B$ are given as follows:

$$A = \begin{bmatrix} 0.0866 & 0.0191 & 0.1394 & 0.1392 & 0.1480 & 0.0241 & 0.0988 & 0.1435 & 0.1883 & 0.1923 \\ 0.1271 & 0.0900 & 0.0924 & 0.0766 & 0.0696 & 0.0466 & 0.1716 & 0.0371 & 0.1097 & 0.0946 \\ 0.0595 & 0.1656 & 0.1953 & 0.1353 & 0.1872 & 0.0587 & 0.0830 & 0.0035 & 0.0215 & 0.0740 \\ 0.1971 & 0.0808 & 0.1301 & 0.0157 & 0.1908 & 0.1505 & 0.0662 & 0.1334 & 0.1394 & 0.1951 \\ 0.0570 & 0.0419 & 0.0470 & 0.0916 & 0.1094 & 0.0640 & 0.0159 & 0.1687 & 0.1224 & 0.0294 \\ 0.1137 & 0.1033 & 0.0379 & 0.0881 & 0.1224 & 0.0139 & 0.0060 & 0.1857 & 0.0732 & 0.0989 \\ 0.1271 & 0.0414 & 0.1232 & 0.1896 & 0.1457 & 0.0997 & 0.1830 & 0.1309 & 0.0673 & 0.0855 \\ 0.0827 & 0.1076 & 0.1498 & 0.1164 & 0.0192 & 0.1888 & 0.1357 & 0.1352 & 0.1086 & 0.1959 \\ 0.1489 & 0.0223 & 0.0018 & 0.0002 & 0.1631 & 0.1272 & 0.0282 & 0.0075 & 0.0351 & 0.0478 \\ 0.0791 & 0.1215 & 0.0219 & 0.1653 & 0.0635 & 0.0230 & 0.1943 & 0.0373 & 0.0253 & 0.1129 \end{bmatrix},$$

$$B = \begin{bmatrix} 0.0728 & 0.0209 & 0.0336 & 0.0269 & 0.0197 & 0.0844 & 0.0358 & 0.0483 & 0.0033 & 0.0559 \\ 0.0459 & 0.0812 & 0.0411 & 0.0338 & 0.0581 & 0.0689 & 0.0084 & 0.0015 & 0.0198 & 0.0053 \\ 0.0136 & 0.0803 & 0.0666 & 0.0570 & 0.0277 & 0.0721 & 0.0613 & 0.0106 & 0.0115 & 0.0699 \\ 0.0848 & 0.0815 & 0.0915 & 0.0766 & 0.0683 & 0.0997 & 0.0389 & 0.0485 & 0.0630 & 0.0102 \\ 0.0858 & 0.0205 & 0.0769 & 0.0968 & 0.0722 & 0.0004 & 0.0201 & 0.0990 & 0.0836 & 0.0750 \\ 0.0168 & 0.0033 & 0.0286 & 0.0740 & 0.0314 & 0.0238 & 0.0183 & 0.0277 & 0.0889 & 0.0123 \\ 0.0795 & 0.0445 & 0.0037 & 0.0776 & 0.0038 & 0.0103 & 0.0183 & 0.0542 & 0.0722 & 0.0544 \\ 0.0399 & 0.0139 & 0.0071 & 0.0227 & 0.0556 & 0.0885 & 0.0062 & 0.0271 & 0.0382 & 0.0641 \\ 0.0142 & 0.0063 & 0.0456 & 0.0536 & 0.0993 & 0.0206 & 0.0264 & 0.0463 & 0.0695 & 0.0907 \\ 0.0301 & 0.0514 & 0.0583 & 0.0007 & 0.0900 & 0.0426 & 0.0385 & 0.0077 & 0.0110 & 0.0930 \end{bmatrix}.$$

For the case $(d, m) = (20, 20)$, we use the same $A$ and $B$ provided in Kim et al. (2021). The result is shown in Figure 3.

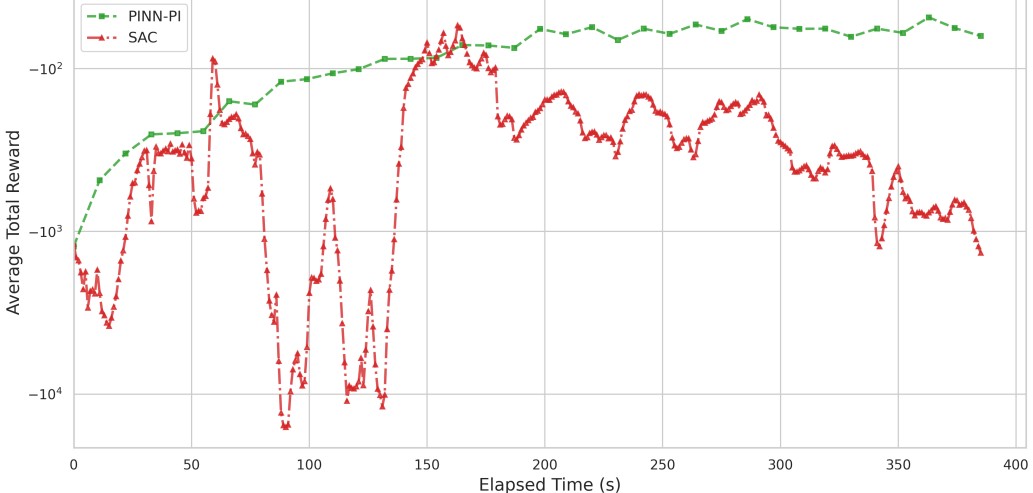

Figure 3: Stochasatic LQR with $(d, m) = (20, 20)$. The green curve (PINN-PI) shows monotonic improvement in evaluation reward, whereas SAC exhibits unstable performance.

For PINN-PI, during each policy evaluation step, we train the value network for 200 gradient steps using the Adam optimizer. The number of collocation points was set to 256, and the learning rate was set to $10^{-4}$ for the case $(d, m) = (5, 5)$, and $10^{-5}$ for the rest cases.

The policy and Q network are optimized using the Adam optimizer with a learning rate of $10^{-3}$.

**Cartpole.**   For policy evaluation in PINN-PI, the value function is trained for 3,000 optimization steps with 64 collocation points using the Adam optimizer with a learning rate of $10^{-3}$.

**Pendulum.**   Similar to the previous task, for policy evaluation in PINN-PI, the value function is trained for 3,000 optimization steps with 64 collocation points using the Adam optimizer with a learning rate of $10^{-3}$.

**Model Architecture.**   All value function approximations $v_n(x; \theta)$ in this work were parameterized by multilayer perceptrons (MLPs) trained to minimize the residual loss in Eq. equation 4. Each MLP consists of 3 hidden layers, with hidden layer widths ranging between 100 and 256 depending on the task. Smooth activation functions such as `tanh` were used to ensure differentiability required for computing PDE residuals.