# OpenReview forum: "Neural Policy Iteration for Stochastic Optimal Control: A Physics-Informed Approach"
_ICLR.cc/2026/Conference — ICLR 2026 Conference Withdrawn Submission_

### Official Review · Reviewer_G72V · 2025-10-31

**Soundness:** 3
**Presentation:** 2
**Contribution:** 3
**Rating:** 6
**Confidence:** 3

**Summary:**

The paper introduces a Physics-Informed Neural Network Policy Iteration (PINN-PI) approach for solving stochastic optimal control problems governed by second-order Hamilton–Jacobi–Bellman (HJB) equations. At each iteration, a neural network approximates the value function by minimizing the residual of a linearized PDE associated with a fixed policy, enabling  $L^2$-error control. The paper establishes theoretical guarantees, including global exponential convergence under classical assumptions and a decomposition of total error into iteration, residual, and policy-mismatch components. Empirical validation on stochastic LQR, pendulum, and cartpole environments demonstrates convergence and stability advantages over model-free baselines.

**Strengths:**

1) The paper is well-written and well-organized and it provides a clear and sound convergence analysis for policy iteration within a physics-informed learning framework. The $L^2$ stability results, Lipschitz continuity of the policy-improvement map, and exponential convergence proofs seems definitely nontrivial.

2)  In contrast to most neural control methods that treat the PDE structure implicitly, this approach explicitly leverages elliptic PDE properties and energy estimates, yielding interpretability and analytical guarantees.

3) The paper's derivation of iteration, residual, and mismatch terms provides valuable insight into how training and approximation errors propagate, a notable strength compared to most PINN-based control papers.

**Weaknesses:**

1) Although PINNs offer mesh-free flexibility, they are still computationally intensive, especially in high-dimensional control spaces. The paper would benefit from reporting computational cost, training time, and scalability comparisons against operator-learning or Galerkin-based solvers.

2) While the theory accounts for residual-based training error, the practical behavior of the neural approximator under finite sampling and stochastic training noise remains untested.

3) The approach assumes full model knowledge (drift, diffusion, and cost functions). While appropriate for theoretical development, it limits applicability to real-world problems with partial or uncertain dynamics.

**Questions:**

1) Could the proposed framework accommodate partial observability or parameter uncertainty, e.g., through learned surrogates for b(x,a)?

2) Is the Lipschitz continuity of the policy-improvement map preserved under learned or non-smooth approximations of the drift?

3) Can the proposed PINN-PI approach be parallelized efficiently across multiple batches to mitigate the cubic scaling of PDE residual computation?

---

### Official Review · Reviewer_mkcP · 2025-11-03

**Soundness:** 2
**Presentation:** 3
**Contribution:** 1
**Rating:** 2
**Confidence:** 4

**Summary:**

The paper proposes a method for learning for the stochastic optimal control via Physics-Informed Neural Network Policy Iteration (PINN-PI). At each iteration, a neural network is trained to approximate the value function by minimizing the residual of a linear
PDE induced by a fixed policy. The authors provide some theoretical groundings to the method via proving the convergence of the algorithm under mild assumptions and validate their approaches with numerical experiments as compared to Soft Actor-Critic (SAC) method in dimensions 5,10,20.

**Strengths:**

The paper has a clear presentation where the readers can easily follow their motivations and they make the algorithm easy to understand. Also, the proposed method, at least to my knowledge, is original. However, I doubt the significance of the method, which I will detail in the Weakness section below.

**Weaknesses:**

I believe the paper has a significant weakness in its empirical evaluations. First, the authors only perform experiments in at most 20 dimensions, which is not generally considered as high in the domain of using deep learning to solve stochastic optimal control (SOC) problems. The authors can refer to the following papers for more challenging experimental settings and baselines:

[1] Hua, Mengjian, Mathieu Laurière, and Eric Vanden-Eijnden. "An Efficient On-Policy Deep Learning Framework for Stochastic Optimal Control." arXiv preprint arXiv:2410.05163 (2024).

[2] Blessing, Denis, et al. "Trust Region Constrained Measure Transport in Path Space for Stochastic Optimal Control and Inference." arXiv preprint arXiv:2508.12511 (2025).

Moreover, it naturally sounds to me that the method is costly especially in high dimensions since the number of collation points used to train on the residual with NN need to be exponentially growing w.r.t. the dimensionality and each policy iteration trains a NN from scratch sounds very expensive. I hope the authors will convince me with more complicated numerical experiments in higher dimensions.

**Questions:**

1. I am wondering if there are any ways to accelerate the training by using the previous trained NN $v_n$ as an initial guess for $v_{n+1}$ in the next policy iteration.

2. In Figure 1 and Figure 2, the two methods start with very different average total rewards and that sounds like a unfair comparison since the two methods should ideally have the same or close initializations. I wonder if SAC would perform much better if it has an initialization as good as the proposed method.

---

### Official Review · Reviewer_gC6Y · 2025-11-05

**Soundness:** 2
**Presentation:** 2
**Contribution:** 2
**Rating:** 2
**Confidence:** 3

**Summary:**

The paper studies the policy iteration for solving stochastic optimal control problems. The authors apply neural network to approximate value function and use it to perform policy update. In theory, the authors analyze the approximation error and the global convergence of policy iterates. Several benchmark problems are used to show the effectiveness of the proposed method.

**Strengths:**

- The neural network-based policy iteration is proposed to solve stochastic optimal control problems. This is a more practical solution since it does not need to solve a PDE explicitly as previous methods.

- The authors characterize the function approximation error and the global convergence of policy iterates. This is a strong theoretical guarantee.

- Experiments demonstrate the outstanding performance of the proposed method, compared to a standard SAC method.

**Weaknesses:**

- Introducing network to policy iteration is not a new idea. It would be helpful if the authors could clarify the key challenges of applying it to stochastic optimal control problems.

- The analysis of approximation error and global convergence is similar as the one in reinforcement learning. It is important to clarify new analysis challenges.

- The neural network-based policy iteration requires accurate model information.

- The provided experiments are limited to textbook examples.

**Questions:**

See comments in Weaknesses.

---

### Note · Authors · 2025-11-12

I have read and agree with the venue's withdrawal policy on behalf of myself and my co-authors.